# Dense Electrode Layers-Supported Microtubular Oxygen Pump

**DOI:** 10.3390/membranes12111114

**Published:** 2022-11-08

**Authors:** Alexey Nikonov, Nikita Pavzderin, Vladimir Khrustov

**Affiliations:** Institute of Electrophysics, Ural Branch, Russian Academy of Sciences, Yekaterinburg 620016, Russia

**Keywords:** oxygen pump, microtubular cell, dense composite electrode layer, thin GDC electrolyte, co-sintering, polarization resistance

## Abstract

An oxygen pump is an electrochemical device that extracts oxygen from the air and has the potential to be used in medicine. The development and test results of a microtubular solid oxide oxygen pump with Ce_0.76_Gd_0.24_O_2−δ_ (GDC) electrolyte are presented. The supporting components of the oxygen pump are symmetrical dense electrode layers made of the La_0.8_Sr_0.2_Co_0.2_Fe_0.8_O_3−δ_ (LSCF)–GDC composite. Studies carried out by impedance spectroscopy on planar samples showed that the polarization resistance of the dense electrodes was greatly lower (by 2.5–5 times) than that of standard porous LSCF electrodes. Microtubular oxygen pumps were fabricated by the isostatic pressing of a stack of tapes rolled into a tube and subsequent co-sintering. The use of a nanosized GDC powder as the initial material for the tapes allowed their co-sintering at 1200 °C. In such a way, the chemical interaction between GDC and LSCF was prevented. Samples of the prepared cells had an outer diameter of 1.9 and 3.5 mm, and the thickness of the electrolyte and of the dense supporting electrodes was 20 and 130 µm, respectively. The specific oxygen productivity of the cells was 0.29 L∙h^−1^∙cm^−2^ at 800 °C and a current density of 1.26 A·cm^−2^ (0.53 V). Thus, the energy consumption with the developed design for the production of 1 L of oxygen was 2.3 W∙h. The microtubular oxygen pumps appeared highly resistant to thermal shock.

## 1. Introduction

Oxygen is one of the most often used chemicals in many industrial processes and in medicine. In the industry, the oxygen is produced by the cryogenic distillation method in efficient optimized plants [1], but the use of this method in medicine is difficult due to logistical and safety problems. Now, oxygen concentrators are used for individual oxygen therapy. The production of 90–95%-purity oxygen in these devices is based on the adsorption of nitrogen by zeolites [2]. The energy consumption of the industrial production of oxygen by pressure swing adsorption (PSA) has been estimated at 0.71 kW∙h∙m^−3^ [3]. However, the energy consumption of commercially available mobile oxygen concentrators varies from 1 to 5 kW∙h∙m^−3^. In addition, oxygen concentrators are quite noisy devices. The noise level of an average concentrator is 40–50 dB. This creates some inconvenience during their long-term use.

Two methods under development based on the oxygen ion conductivity of ceramic membranes allow obtaining high-purity oxygen (with a purity of more than 99%) [2,4,5]. High operating temperatures (600–900 °C) are required in both methods to achieve a suitable level of ionic conductivity. In the first method, dense layers of mixed ion–electron materials, often referred to as ion transport membranes (ITM), are used to separate the oxygen from the air. Oxygen in the ion form flows across the membrane layer, while electrons move in the opposite direction completing the internal electronic circuit. In this case, the driving force of the oxygen ions flow is the difference in oxygen partial pressures in different sides of the membrane. The pressure on the air feed side is typically 7–20 atm, and the pressure on the on the permeate side varies from low to sub-atmospheric levels [4]. The second method is essentially based on the electrolysis of oxygen from the air. Oxygen under the action of an external electric field passes through the crystal lattice of a dense ion-conducting oxide material (electrolyte) from the cathode to the anode. Electrochemical devices that work according to this principle are called oxygen pumps (OP). They are widely used in laboratory practice to create and maintain atmospheres with a given oxygen partial pressure [6,7].

The characteristics of the described oxygen production technologies are presented in Table 1. The theoretical estimate of the minimum energy consumption for oxygen production by the electrochemical method is ~0.19 kW∙h∙m^−3^ at 900 °C [8]. In addition, the absence of compressors required for the PSA makes this technology silent. Thus, OP is a promising technology for oxygen production for individual oxygen therapy. However, oxygen pumps have still not been adapted to the medical use due to their long start-up time. The most common electrolyte material of Y_2_O_3_-stabilized ZrO_2_ (YSZ) has a suitable level of ionic conductivity only at temperatures above 800 °C. Therefore, in order to produce oxygen, the oxygen pump must be heated to the operating temperature, which takes several hours. Rapid heating results in the destruction of the electrochemical part, since its multilayer structure, consisting of dissimilar ceramic materials, does not withstand a thermal shock.

When developing another high-temperature electrochemical device of a solid oxide fuel cell (SOFC), it was shown that the microtubular cell design survives a thermal shock, and the heating time is reduced to several seconds [9,10]. However, works devoted to the microtubular oxygen pump have not been published. In general, there are very few investigations of OP based on solid oxide electrolytes for oxygen production [5,8,11,12,13,14,15]. In the vast majority of works [8,11,12,13,14], electrolyte-supported oxygen pumps are considered. In this case, a thick electrolyte layer causes a high internal resistance of the OP, since the specific conductivity of solid electrolytes is 2–3 orders of magnitude lower than that of electrode materials [16]. Therefore, mainly, the electrolyte layer resistance should be reduced to minimize the OP ohmic resistance. This can be achieved in two ways. Firstly, an electrolyte with a conductivity higher than that of YSZ has to be used. In particular, Gd_2_O_3_-or Sm_2_O_3_-doped CeO_2_ (GDC or SmDC) is such an electrolyte material [17,18]. Secondly, the electrolyte layer thickness has to be reduced, resulting in a design change of the cell to an electrode-supported one. Herewith, an important factor is the preservation of the electrolyte layer gasproof (continuity), since this layer separates the cathode and anode gas spaces.

Despite the fact that electrode processes have been widely studied during SOFC development, there are still only general recommendations for minimizing polarization losses, since the processes depend on many factors. Nevertheless, it is generally recognized that in high-temperature electrochemical devices, the electrode reactions occur at the three-phase boundary where the electrolyte, electrode, and gas meet [7,16]. Therefore, to reduce the polarization resistance, it is possible to try to expand the three-phase boundary by using composites and nanostructured electrodes [19,20]. Recently, it was shown that the introduction of a dense layer with mixed ionic–electronic conduction on the electrolyte–porous electrode interface resulted in a decrease of polarization losses [21,22,23,24,25]. In this works, systems with various cathode and electrolyte materials were studied, which makes it difficult to identify general regularities. Moreover, the thicknesses of the investigated dense electrode layers were limited to a few microns. In [25], it was suggested that the effect of the dense layer on the electrode polarization resistance depends on the ionic component of its conductivity. Therefore, the use of a mixed conductor with a high ionic conductivity should allow the creation of a dense layer that, at the same time, will not worsen the electrode performance and will has sufficient thickness to take on the role of a supporting component of a cell. In addition, dense electrode layers along with the electrolyte will separate the anode and cathode gas spaces, increasing the design reliability. However, the ionic conductivity of mixed conductors with a perovskite structure correlates with their thermal expansion [26]. With an increase of ionic conductivity, the thermal expansion coefficient (TEC) of materials also increases. The mismatch of the thermal expansion of the electrode and electrolyte materials results in a breakdown of the cell upon heating. The simplest method for increasing the ionic component of the conductivity of a dense electrode layer is the use of a composite [27] which additionally conforms the TEC of the functional materials [28].

In this work, the effect of a dense composite layer on the electrode characteristics was investigated, and the performance of a microtubular oxygen pump with supporting dense electrode layers was determined.

## 2. Materials and Methods

A nanopowder of the electrolyte material Ce_0.73_Gd_0.27_O_2−δ_ (GDC) was prepared by laser evaporation in IEP UB RAS (Yekaterinburg, Russia) [29]. The powder of the electrode material La_0.8_Sr_0.2_Co_0.2_Fe_0.8_O_3−δ_ (LSCF) was synthesized by solution combustion. The chemical agents used were La_2_O_3_, SrCO_3_, Fe(NO_3_)_3_ (purity > 98%) and Co(NO_3_)_2_ (purity > 99%) (JSC Vekton, Yekaterinburg, Russia). The required quantities of La_2_O_3_ and SrCO_3_ were dissolved in a 0.1 N solution of HNO_3_, Fe(NO_3_)_3_, and Co(NO_3_)_2_ was dissolved in water. The combustible organic substances used were glycine (AMK Ltd., St. Petersburg, Russia) and citric acid (purity > 98%) (Weifang Ensigh Industry Co., Ltd., Weifang, China) in amounts which ensured the carrying out of the combustion reaction in the area of reductive combustion. The reaction products were ground and annealed in stages at temperatures of 850, 950, and 1100 °C to remove the organic phase residues and form a crystal structure. The phase composition and the specific surface area of the synthesized powders were determined by the diffractometer D8 Discover and the analyzer TriStar 3000, respectively. The X-ray data were analyzed using TOPAS 3 software.

The composites were prepared from LSCF and GDC-10 powders at weight ratios of 1:1 and 2:3. Hereinafter, the composites are designated as LC/C_X/Y, where X and Y are the mass fractions of LSCF and GDC, respectively. To provide the composites homogeneity, the powders were thoroughly mixed by using the disperser UZG8-0.4/22 and a laboratory gravity mixer for 2 full days

The sintering kinetics and thermal expansion of the materials were studied by a Dil 402C dilatometer in air atmosphere. The heating rate was 5 °C/min. The shrinkage samples had the form of disks with 8 mm diameter and ~3 mm thickness, which were pressed to a relative density of ~0.5–0.6. The linear expansion measurements of the materials were carried out on bar-shaped samples sintered to a density closed to theoretical one. The density of the samples was determined by hydrostatic weighing.

Co-sintering of GDC with the electrode materials was examined on disk samples with an electrode/electrolyte structure, made by using tapes. The tapes of electrolyte and electrode materials were cast from a slurry consisting of 82.1 wt.% powder, 14 wt.% polyvinyl butyral (PVB) and 3.9 wt.% triethylene glycol dimethacrylate. The solvent was isopropanol. From the prepared tapes, disks with a 15 mm diameter were cut out, and stacks from them with a specified number of layers were pressed at a pressure of 300 MPa. The sample co-sintering was investigated by a V-600 cathetometer (Lascar Electronics, Erie, PA, USA) and a SONY DSC-V3 digital camera (Tokyo, Japan). During the test, the sample was placed in a special oven with a window on a smooth ceramic substrate mounted directly on the control thermocouple.

After the implementation of co-sintering, disk samples with an electrode-electrolyte-electrode structure were fabricated. The symmetrical samples were used to study the effect of dense composite layers on the electrode characteristics according to the method described in [25]. In the first stage, the samples with electrodes consisting only of a dense composite layer (DE) were studied. The thickness of the dense electrode layers was varied from 15 to 150 μm. In the second stage, double-layer electrodes consisting of a dense composite layer and a porous layer (DE + PE) were studied. The porous LSCF layers were formed over the already tested DE by painting and subsequent firing at 1100 °C. The slurry for applying the porous layer was made by mixing the LSCF powder with 5 wt.% PVB. A sample with only a porous LSCF electrode (PE) was also produced for comparison. The polarization resistance of the electrodes was investigated by impedance spectroscopy using a Solartron Sl-1287/1260 (Solartron Public Co., Ltd., Hampshire, UK). The impedance spectra were recorded at the voltage of 15 mV in the frequency range 0.1 Hz–0.5 MHz.

Three electrochemical cells with different thicknesses of the functional layers for research in the oxygen pump mode were assembled based on the obtained symmetrical samples. Porous LSCF electrodes were deposited on both surfaces of the samples and fired at 1100 °C for 1 h. Platinum wire probes were placed over the porous electrodes, and a platinum paste was used to ensure a good electrical contact. The samples were glued by a glass sealant to the YSZ tube that separated the gas spaces. The current–voltage performances of the cells were measured using a P-40X potentiostat/galvanostat (Electrochemical Instruments, Dubai, United Arab Emirates) combined with an impedance measurement module. Oxygen flow generated by the planar cells was measured using the displacement technique. The oxygen permeation flux at the passing of a certain current through the cell was determined using a soap bubble meter.

Microtubular oxygen pump samples were fabricated from GDC and LC/C_2/3 tapes. A green microtubular sample was formed by the serial winding of the functional tapes with the required layer thickness on a hard metal rod. Rods with diameters of 1.5 and 4 mm were used. Compaction (lamination) of the layers was performed in a liquid isostat at a pressure of 200 MPa. After the removal of the rod, the sample was sintered. The sintering mode was chosen based on the results of the co-sintering of planar samples. To study the microtubular sample performances, platinum wire probes were installed inside and outside the tube. The electrical contact between the probes and the dense composite layers was provided by applying the LSCF slurry over the probes. The LSCF layer was fired at 1100 °C. Using a glass sealant, the microtubular OP was sealed at one end to the YSZ plate and at the other end to the YSZ tube that collected the oxygen and moved it to the cold zone. A study of the microtubular OP performance was carried out using SI-1287/1260. The measurements were performed at temperatures of 700, 750, and 800 °C. The impedance spectra were recorded at the voltage of 15 mV in the frequency range 0.1 Hz–0.5 MHz. Current–voltage performances were taken up to a voltage of 0.9 V. The oxygen permeation flux was determined by the same technique as for the planar samples. The resistance of microtubular OP to thermal cycling was investigated by installing the cell in the hot oven zone at a temperature of 800 °C and then moving it to the zone at a temperature of 400 °C. The duration of the mechanical extraction/installation of the cell was 1–2 s. The temperature measurement on the OP sample showed that it took to reach 800 °C, when the cell was placed on the hot zone, or 400 °C when it was removed from it. The microtubular sample was subjected to thermal cycling 25 times. The device operating in the potentiostatic mode at 0.5 V set the current during the tests.

## 3. Results

### 3.1. Implementation of Co-Sintering of the LSCF–GDC Composite and the GDC Layer

The obtained GDC and LSCF powders were single-phase. GDC had a cubic fluorite type structure with the lattice parameter a = 5.424 Å. LSCF had a perovskite type structure (space group R-3c) with the lattice parameters a = 5.501 and c = 13.407 Å. The XRD data and the results of the study of the interaction of GDC with LSCF at temperatures of 1200 and 1300 °C are presented in detail elsewhere [25]. According to the BET data, the LSCF particle size was 0.31 µm, while the GDC powder consisted of sphere-like shape particles with an average size of 10 nm. To change the sintering kinetics, the initial GDC powder was annealed for 4 h at temperatures of 1000 °C (GDC-10), 1100 °C (GDC-11) and 1200 °C (GDC-12). The average particle sizes of the GDC-10, GDC-11 and GDC-12 powders were 44, 109 and 328 nm, respectively.

The sintering kinetics of the initial materials of GDC and LSCF were significantly different (Figure 1). First, GDC started shrinking earlier at 450 °C than LSCF due to the much smaller size of the GDC particles. Secondly, LSCF possessed a higher sintering rate. Annealing of the GDC powder resulting in an increase of its particles size allowed the start temperatures of the sintering of the electrolyte material and LSCF to become closer. It was observed that the sintering kinetics of GDC-11 was the closest to that of LSCF (Figure 1). Co-sintering studies carried out with using a cathetometer showed that at a temperature of about 1250 °C, the tape samples delaminated due to a difference in the shrinkage rates of LSCF and GDC in the high-temperature region. Samples without visible defects were obtained by limiting the heating temperature to 1200 °C. However, during cooling (in the region of 400–300 °C), they cracked, which was most likely caused by the difference in the TEC of LSCF and GDC (Table 2). The results of the study of the sintering kinetics and TEC of the LC/C_1/1 composite are presented in Figure 1 and in Table 2, respectively. It can be seen that the TEC values of LC/C_1/1 and GDC TEC were significantly closer than the TEC values of LSCF and GDC. The shrinkage of the composite, although similar, did not perfectly match the shrinkage of GDC-10. Therefore, to implement the co-sintering, a composite with a higher content of the electrolyte material LC/C_2/3 was used. Of course, a further increase of the GDC-10 content in the composite would result in a better thermomechanical compatibility of the electrode and electrolyte layers. However, this approach is not applicable in case of electrodes. The conductivity of the composite was shown to already drop by a factor of 7 at a ratio of LSCF/GDC of 2:3 [30], and when the content of LSCF is below the percolation threshold (about 30 vol.%), the composite layer cannot play the role of an electrode.

Based on the performed studies, the following temperature regime of co-sintering was chosen: slow heating (1 °C/min) to 600 °C to remove polymer additives, heating at a rate of 5 °C/min to 1200 °C and subsequent exposure of 10 h. Double-layer samples of LC/C_2/3//GDC-10 during sintering underwent a series of deformations (Figure 2) caused by different shrinkage rates of the materials at different temperatures. Although the double-layer samples were curved after sintering, there was no lamination. Symmetrical samples of LC/C_2/3//GDC-10//LC/C_2/3 and LC/C_1/1//GDC-10//LC/C_1/1 that were used later for impedance measurements and research in the oxygen pump mode, respectively, were fabricated under the same conditions. After co-sintering, the triple-layer samples were not completely flat, but their deformation was significantly less than that of double-layer samples. The use of a tightening weight solved this problem. Figure 3 shows images of cross fractures of LC/C_1/1//GDC//LC/C_1/1 samples with different thicknesses of the functional layers, taken with an Olympus BX41M-LED optical microscope.

### 3.2. Influence of a Dense Electrode Layer on the Polarization Resistance

Figure 4 shows the impedance spectra of a number of electrodes with different structures: PE—porous electrode LSCF, DE—dense electrode LC/C_2/3 and DE + PE—double-layer electrode, consisting of a dense and a porous layer. In the legend of Figure 4, the thickness of the dense composite layer is indicated in parentheses. In each spectrum, two parts can be distinguished: a high-frequency (HF) portion, which according to the literature [31] corresponds to the charge transfer process across the electrolyte–electrode interface, and a low-frequency (LF) portion, which corresponds to the processes occurring on the mixed ionic–electronic conductor–gas interface (adsorption/desorption of oxygen, dissociation of oxygen, diffusion over the surface and the relationship of these processes) [31]. The analysis of the impedance spectra was carried out according to an equivalent circuit presented in the inset of Figure 4. The circuit consisted of series-connected resistances R_s_ and two elements (RQ). The resistance R_s_ corresponds to ohmic losses and is mainly determined by the electrolyte resistance. In Figure 4, the beginning of the impedance spectrum is artificially aligned for ease of comparison. The resistances R_1_ and R_2_ correspond to the double values of the high- and low-frequency contributions to the polarization resistance (R_η-HF_ and R_η-LF_).

The results obtained after dividing the obtained impedance spectra into high-(R_η-HF_) and low-frequency (R_η-LF_) parts are shown in Figure 5. The dependence of the total polarization resistance of the electrodes with the DE and DE + PE structures on the dense composite layer thickness is presented in Figure 6. It can be seen that all double-layer electrodes had a significantly lower (by a factor of 2.5–5) polarization resistance compared to the single-layer porous LSCF electrode. Herewith, the improvement of the electrode characteristics upon the introduction of a dense layer was associated both with the process of oxygen ions transfer through the electrolyte–electrode interface and with the processes occurring at the electrode–gas interface. The electrode with a dense layer thickness of 15 µm was the only exception. This electrode had a 2.5-time higher R_η-LF_ than the porous electrode and a 2–3-time higher R_η-HF_ than the other DE electrodes. The formation of a porous electrode over the DE (15 μm) layers resulted in a decrease of the high-frequency contribution of the polarization resistance up to the level of other DE electrodes. The R_η-LF_ value of the DE + PE (15 μm) electrode was the same as that of the PE electrode but still ~2.5 times higher than that of the other DE + PE electrodes. It should be noted that the characteristics of the single- and double-layer electrodes were identical when the dense layer thickness was more than 45 μm.

### 3.3. Characteristics of the Planar Oxygen Pump

The planar electrochemical cells for research in the oxygen pump mode were assembled based on the LC/C_1/1//GDC-10//LC/C_1/1 samples, as well as on the PE//GDC-10//PE sample without dense electrode layers. Table 3 shows the thicknesses of the functional layers of the planar cells.

The current–voltage performances of the cells at 800 °C are presented in Figure 7. The internal resistance of the OP-3 cell with dense electrode layers was lower by ~1.9 times than the resistance of the OP-2 cell and by ~6.8 times than that of the OP-1 cell. A comparison of the impedance spectra of the planar oxygen pumps (Figure 8) showed that the transition from a supporting electrolyte to a supporting dense electrode layer resulted both in a decrease in the ohmic resistance of the cells (R_s_) and in a decrease in the polarization resistance (R_η_). The ohmic resistance of OP-1 and OP-3 differed by a factor of ~11 due to the high conductivity of the LSCF/GDC composite which was 2–3 orders of magnitude higher than that of the GDC electrolyte [30]. The polarization resistance of OP-1 was ~4 times higher than that of OP-3, which agreed with the data presented in Section 3.2. Thus, the transfer of a mechanical supporting function in a high-temperature oxygen pump to dense electrode layers allowed significantly reducing its internal resistance and thereby increasing oxygen production with the same energy consumption.

### 3.4. Characteristics of the Microtubular Oxygen Pump

Microtubular samples of the oxygen pump were sintered in the same temperature regime as the planar OP. Defect-free samples with outer diameters of 1.9 and 3.5 mm and lengths of the order of 24 and 15 mm, respectively, were obtained (Figure 9a). The microtubular samples with thick supporting dense electrode layers of 80 to 150 µm thickness and thin GDC electrolyte layers of 20–40 µm thickness (Figure 9b) were fabricated by varying the tape layers number. The studies were carried out on samples with supporting layer and electrolyte layer with thicknesses of 130 and 20 µm, respectively.

Figure 10 shows the current–voltage performances of the microtubular oxygen pump at various temperatures. The internal resistance of the oxygen pump increased by 1.5 times (from 0.49 to 0.73 Ohm) at a decreasing operating temperature from 800 to 700 °C. The non-linear shape of the curves was due both to the non-linear dependence of the electrode polarization resistance and a decrease in the electrolyte resistance because of heating the sample when high currents flowed. The temperature of the oxygen pump at a current density of 2 A∙cm^−2^ increased from 800 to 828 °C. At 800 °C, and at a current density of 1.26 A∙cm^−2^ (0.53 V), the specific oxygen flow rate of the cell was 0.29 L∙h^−1^∙cm^−2^. Thus, the energy consumption for the production of 1 L of oxygen with the developed design was 2.3 W∙h (Figure 11).

The results of thermal cycling of the microtubular oxygen pump are shown in Figure 12. It was observed that the current density flowing through the OP at 800 °C and applied constant voltage of 0.5 V remained unchanged from cycle to cycle, which indicated the absence of any changes in the structure of the oxygen pump and the high resistance of the microtubular design to thermal shock.

## 4. Discussion

During the study of electrodes with dense layers less than 1 µm thick [21,22,23,24], it was shown that the improvement of the electrode characteristics was due to processes at the electrolyte–electrode interface. This was apparently because the transfer of oxygen ions occurred over the entire electrode–electrolyte interface when a dense layer with mixed ion–electron conductivity was present in the electrode structure. In this case, oxygen ions only jumped from one crystal lattice to another. In a porous electrode, pores occupy part of the electrode–electrolyte interface, so that the area of oxygen transfer is effectively less. However, the numerical data of different works [21,22,23,24] differ. In [22,23], the polarization resistance of a double-layer electrode consisting of a porous layer and a dense layer was ~30% lower than that of a porous electrode. In contrast, in [21], the polarization resistance of the electrode dropped by three times when a dense electrode layer was introduced onto the electrolyte surface. In [25], it was shown that in samples with dense layers that were several microns thick, the effect of the dense layer on the polarization resistance correlated with ASR = ρ∙δ of the dense layer (where ρ and δ are the material resistivity and the thickness of the dense layer, respectively). After a certain value, an increase of ASR resulted in a deterioration of the electrode characteristics, i.e., a dense electrode layer made of a material with a certain specific conductivity (1/ρ) could both reduce and increase the polarization resistance of a double-layer electrode, depending on its thickness. Herewith, it appeared that the ionic component of conductivity of the mixed conductor is more important that its total conductivity.

The last suggestion was confirmed by the data presented in this paper. The GDC electrolyte material has an oxygen ion conductivity that is an order of magnitude higher (0.032 S cm^−1^ at 800 °C [30]) than that of La_0.8_Sr_0.2_Co_0.2_Fe_0.8_O_3−δ_ (0.0023 S cm^−1^ at 800 °C [26]). Consequently, composites based on these materials with LSCF-to-GDC ratios of 1:1 and 2:3 also have ionic conductivity several times higher than that of pure LSCF. Due to the high ionic conductivity of the composite material, the dense electrode layer can be thick enough to take on the role of the supporting component without increasing the polarization resistance of the electrode. In addition, as shown above, the introduction of the dense composite layer into the electrode structure resulted both in the facilitation of the oxygen ions transfer across the electrolyte–electrode interface and in the acceleration of the processes at the electrode–gas interface.

It is known that the oxygen reduction reaction can proceed not only near the triple-phase boundary (TPB) but also over the entire surface of the cathode material if it possess-es mixed ion–electron conductivity (MIEC) (LSCF is such material) [27,28]. Therefore, in the case of a dense MIEC cathode, the reaction has to take place on its surface facing the gas even without any porous layer. The formation of a porous layer over a dense one certainly results in an expansion of the reaction region and, consequently, in a decrease of the polarization resistance [25]. However, the use of a composite as a dense electrode layer significantly changes the situation. It was shown [32,33] that the coating of LSCF with SmDC particles resulted in an increase of a factor of 10 in the oxygen surface exchange rate. Thus, the lower value of the low-frequency contribution of the polarization resistance of dense composite electrodes compared to that of a porous LSCF electrode (Figure 5b) was due to precisely to this phenomenon. The effect of GDC introduction on the reaction rate at the dense electrode (DE)–gas interface was so great that the expansion of the reaction region due to the formation of a porous layer (DE + PE) was not noticeable. Only in the case of a 15 µm-thick dense electrode with a low surface exchange rate, coating with the porous layer had a significant positive effect (Figure 6). The reason of the high value of R_η-LF_ of this dense electrode is currently not clear and will require additional research.

The theoretical calculation of the minimum energy consumption for oxygen production by the electrochemical method was performed under the assumption that energy is consumed only to overcome the EMF arising due to the difference in oxygen partial pressures (0.21 from the air feed side and 1 from the pure oxygen collection side) without taking into account OP internal losses [8]. The EMF was determined in accordance with the Nernst equation. The estimated minimum energy consumption in the temperature range of 700–900 °C varied from 0.16 to 0.19 kW∙h∙m^−3^ [8]. However, the OP operating voltage (*U*) has to be higher than the Nernst EMF (*E_OCV_*) and is defined as:*U* = *E_OCV_* + *I*∙*R_in_*(1)
where *I* is the current flowing through OP, and *R_in_* is the OP internal resistance consisting of ohmic and polarization losses. A voltage increase inevitably results in an energy consumption increase (*W* = *U*∙*I*). Thus, the minimization of the OP internal losses is a necessary condition for the competitive ability of the electrochemical technology compared with other technologies of oxygen production (Table 1).

A decrease in the OP operating temperature results in a slight decrease of *E_OCV_*, which could have a positive effect on energy consumption. However, as the temperature decreases, the solid electrolytes resistivity and the polarization resistance of the electrodes increase exponentially [16,17]. Therefore, a decrease in the OP operating temperature results in an energy consumption increase.

As mentioned in the Introduction, the electrolyte layer provides the main contribution to the oxygen pump ohmic resistance. A reduction in the solid electrolyte layer thickness is an effective way to reduce of its resistance. This could be confirmed by the data presented above (Figure 7 and Figure 8). However, the overwhelming majority of works on the development of OP based on solid electrolytes have been devoted to designs with a supporting electrolyte [8,11,12,13,14]. Moreover, in [5,8,13,14], small laboratory samples of planar-design OP were studied. The characteristics of the OP stacks were reported only for tubular cells [8,12], which was apparently due to the convenience of separating air and oxygen gas spaces in tubular geometry. The energy consumptions of experimental samples of oxygen pumps were 6.5 kW∙h∙m^−3^ at 660 °C [8], about 5.5 kW∙h∙m^−3^ at 800 °C [12], 11.9 kW∙h∙m^−3^ at 800 °C [13] and 6.1 kW∙h∙m^−3^ at ~750 °C [14]. Thus, at the moment, the microtubular oxygen pump with supporting dense composite electrodes designed and presented in this paper has the lowest internal resistance and, therefore, has the lowest energy consumption for oxygen production (2.3 kW∙h∙m^−3^).

## 5. Conclusions

In this work, the fabrication method of a microtubular oxygen pump with a new structure consisting of functional layers was developed, and the characteristics of the obtained OP were studied. The principal feature of the structure is the presence of dense electrode layers obtained from a composite based on an LSCF mixed ion–electrode conductor and a GDC electrolyte. The dense layers are the supporting element of the OP that allowed reducing the electrolyte layer thickness to 20 μm. In addition, the dense electrode layers along with the electrolyte separate the cathode and anode gas spaces, thereby increasing the system reliability. It was shown that the introduction of dense composite layers in the electrode structure resulted in a significant decrease in the polarization resistance. Thus, a low internal resistance of the microtubular oxygen pump and, consequently, la ow energy consumption for oxygen production (2.3 kW∙h∙m^−3^ at 800 °C) were achieved by the new structure containing dense composite electrode layers.

Composites are often used in the creation of high-temperature electrochemical devices based on solid electrolytes to match the TEC of different functional layers. It is possible that the double-layer structure of electrodes with a dense composite layer will find application not only in oxygen pumps, but also in SOFCs and electrolyzers.

## Figures and Tables

**Figure 1 membranes-12-01114-f001:**
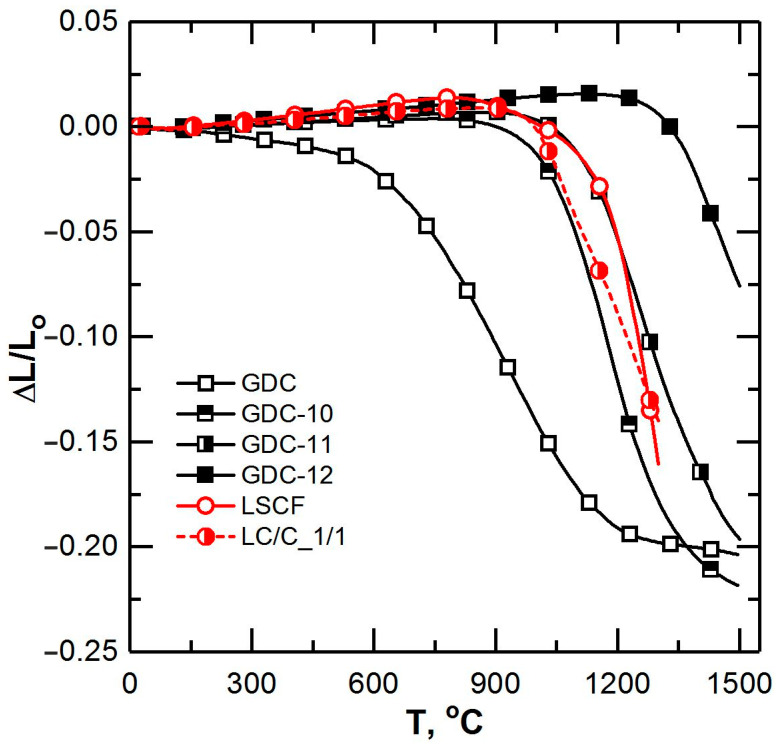
Linear shrinkage curves for the investigated materials.

**Figure 2 membranes-12-01114-f002:**
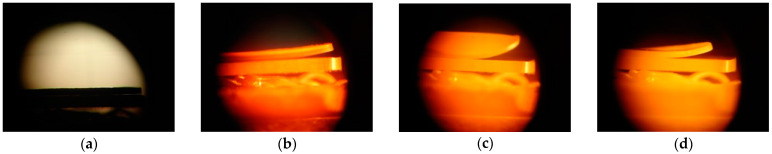
Image of a double-layer sample of LC/C_2/3//GDC-10 during heating at (**a**) 20 °C, (**b**) 915 °C, (**c**) 1120 °C and (**d**) 1200 °C. The LC/C_2/3 layer is at the bottom; the GDC-10 layer is at the top.

**Figure 3 membranes-12-01114-f003:**
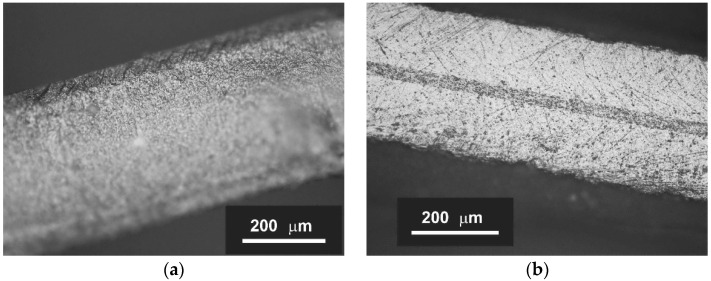
Microphotographs of cross fractures of symmetric LC/C_1/1//GDC//LC/C_1/1 samples with different thicknesses of the functional layers: (**a**) supporting electrolyte, (**b**) supporting dense electrode layers.

**Figure 4 membranes-12-01114-f004:**
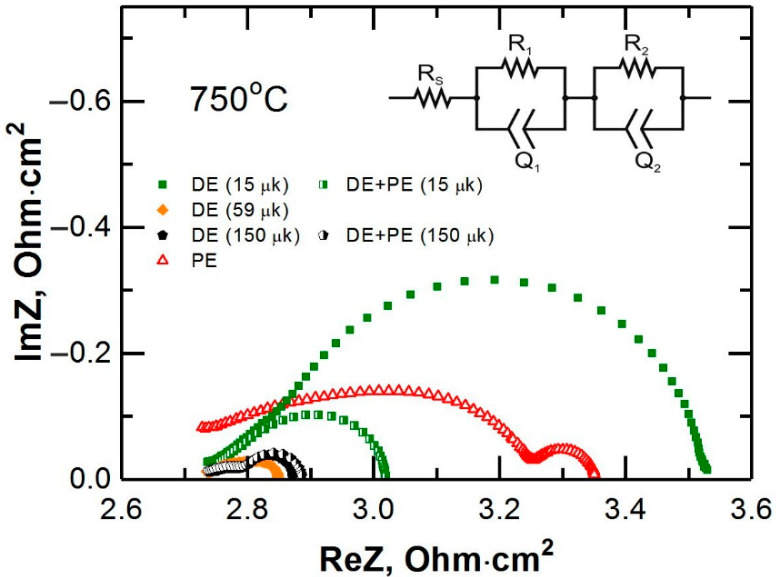
Impedance spectra of the electrodes with different structures at 750 °C. Inset: equivalent circuit.

**Figure 5 membranes-12-01114-f005:**
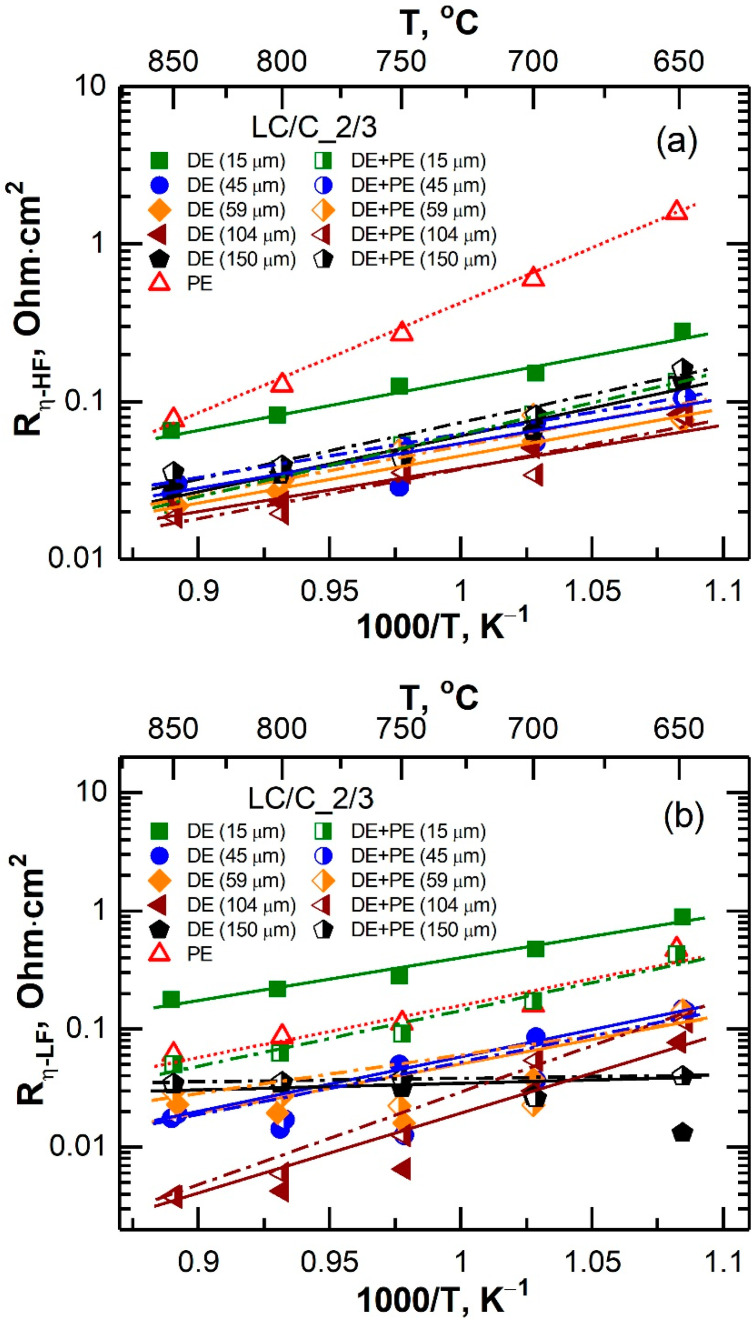
Temperature dependence of the (**a**) high-frequency (HF) and (**b**) low-frequency (LF) contribution of the polarization resistance of the investigated electrodes.

**Figure 6 membranes-12-01114-f006:**
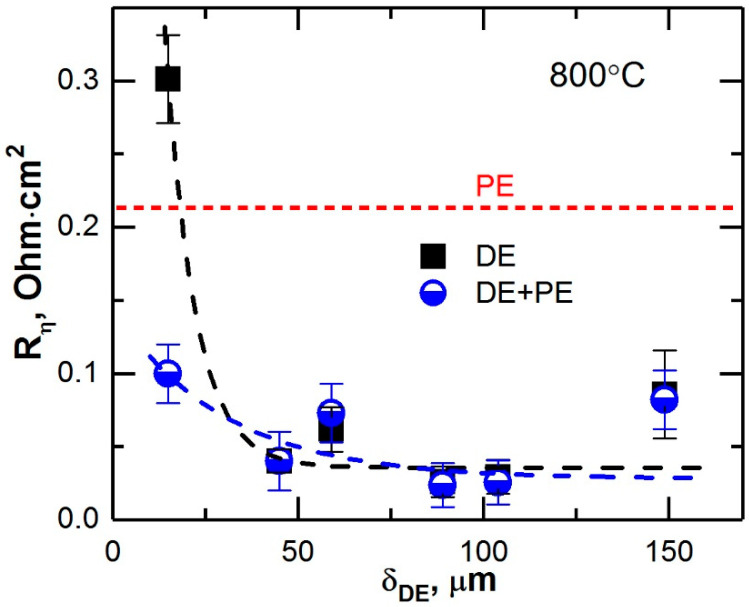
Dependence of the polarization resistance of the electrodes with DE and DE + PE structures on the dense layer thickness.

**Figure 7 membranes-12-01114-f007:**
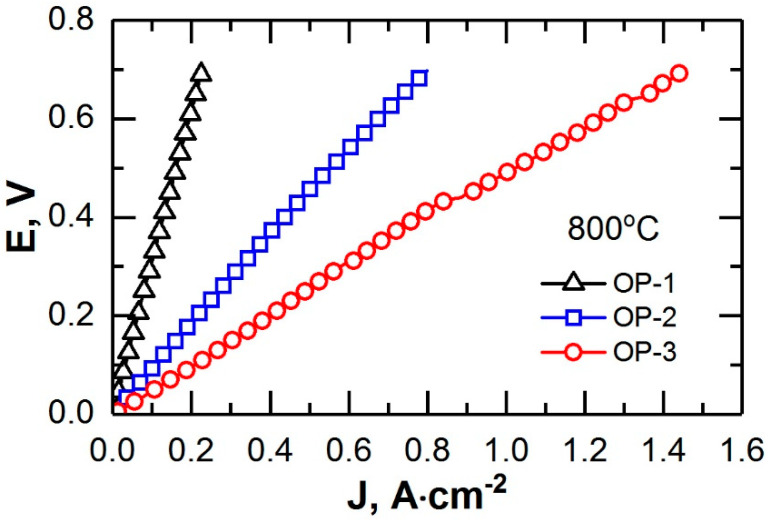
Current–voltage performances of the planar oxygen pumps at 800 °C.

**Figure 8 membranes-12-01114-f008:**
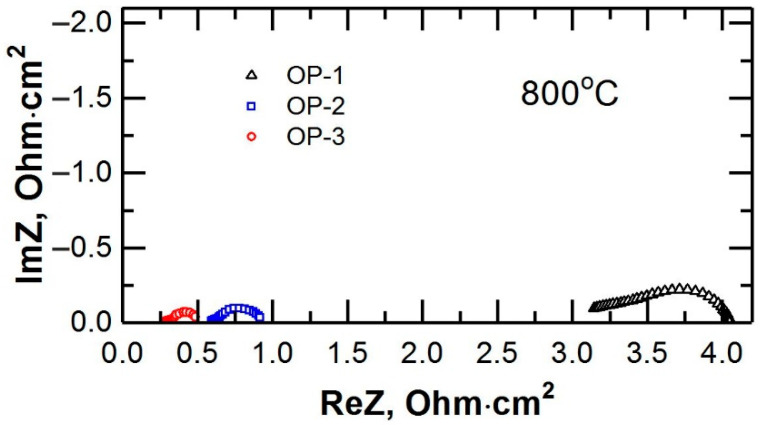
Impedance spectra of the planar oxygen pumps at 800 °C.

**Figure 9 membranes-12-01114-f009:**
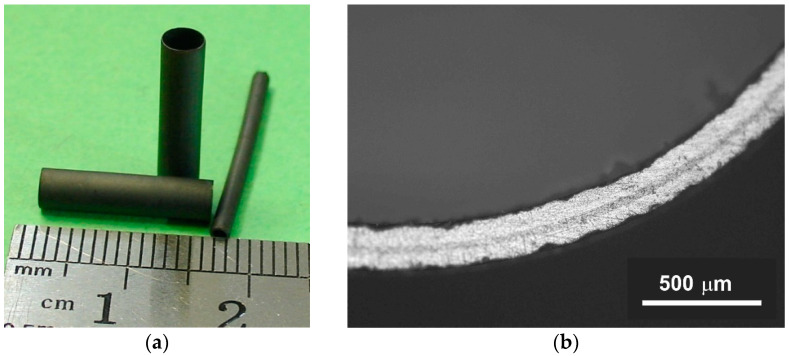
Microtubular oxygen pumps samples: (**a**) appearance and (**b**) microphotographs of the polished cross section.

**Figure 10 membranes-12-01114-f010:**
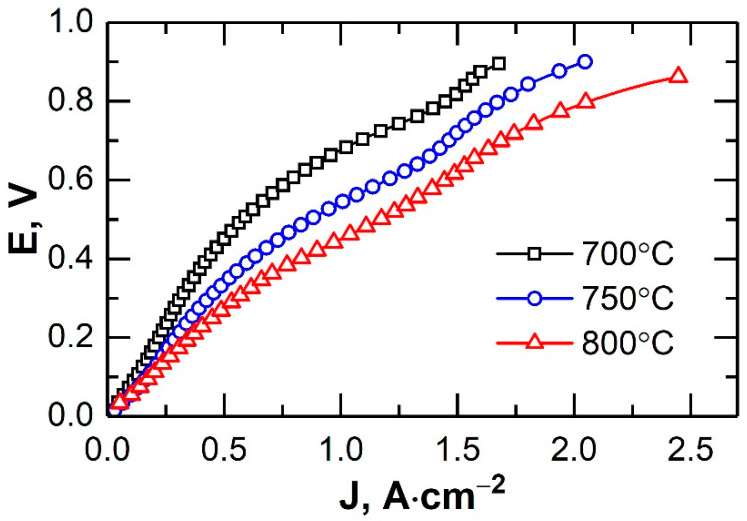
Current–voltage performance of the microtubular oxygen pump.

**Figure 11 membranes-12-01114-f011:**
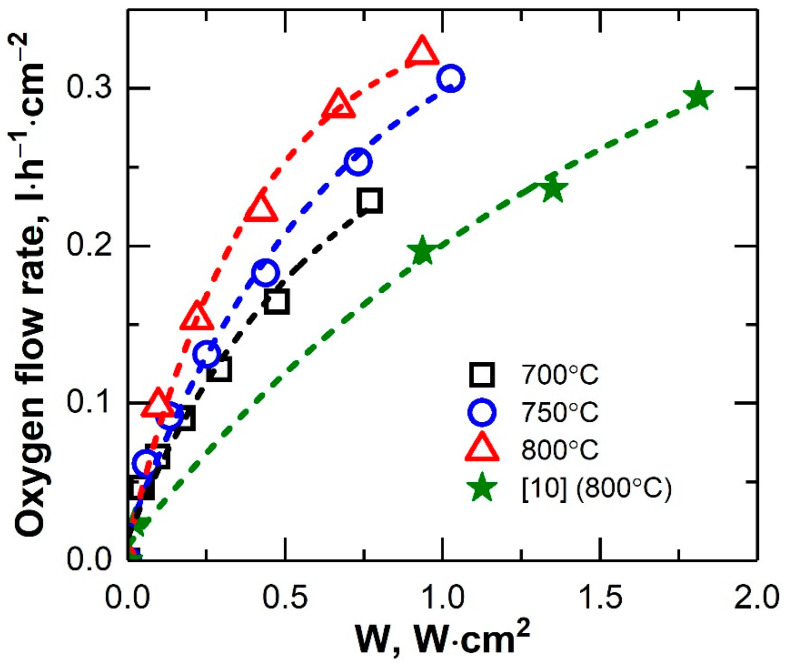
Dependence of the specific oxygen flow rate of the microtubular oxygen pump on the specific applied power.

**Figure 12 membranes-12-01114-f012:**
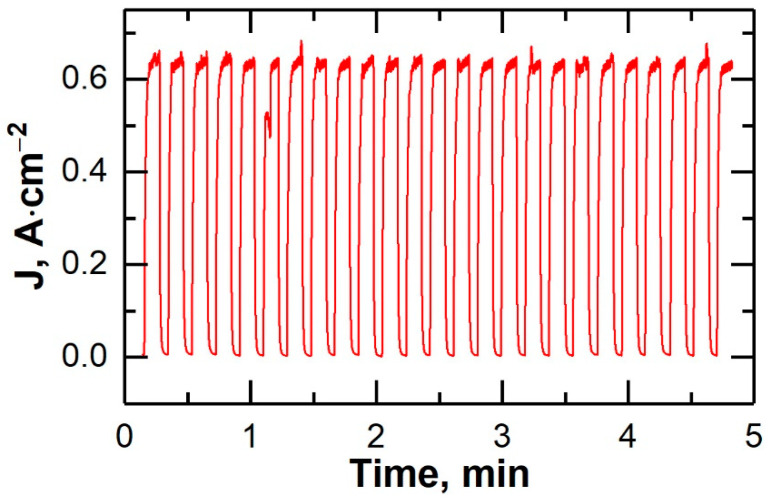
Thermal cycling performance of the microtubular oxygen pump between 400 and 800 °C at the voltage of 0.5 V.

**Table 1 membranes-12-01114-t001:** Comparison of oxygen production technologies.

Technology	Development Stage	O_2_ Purity, %	Capacity, Tons/Day	Energy Consumption, kW∙h∙m^−3^	Mobility	Reference
Cryogenic	Matured	99.5	up to 4000	0.29	No	[3]
Adsorption	Matured	90–95	up to 300	0.71	Yes	[3]
Membrane (ITM)	R & D phase	99.9	laboratory scale	0.57	Yes	[3]
Electrochemical (OP)	R & D phase	99.9	laboratory scale	0.19 *	Yes	[8]

*—minimum theoretical estimate for the electrochemical cell.

**Table 2 membranes-12-01114-t002:** TEC of the investigated materials.

Composition	ΔT, °C	TEC, 10^−6^∙K^−1^
LSCF	100–815	14.5 ± 0.1
	815–1200	20.6 ± 0.2
LC/C_1/1	100–600	10.0 ± 0.1
	600–1200	15.0 ± 0.1
GDC	100–1200	12.7 ± 0.02

**Table 3 membranes-12-01114-t003:** Structure of symmetrical samples studied in the oxygen pump mode.

Planar Cell Designation	Structure of OP			δ_DE_, mkm	δ_GDC_, mkm
	PE	DE	Electrolyte		
OP-1	LSCF	-	GDC-10	-	440
OP-2	LSCF	LC/C_1/1	GDC-10	30	280
OP-3	LSCF	LC/C_1/1	GDC-10	140	35

## Data Availability

Not applicable.

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
