# Peer review of "Dense Electrode Layers-Supported Microtubular Oxygen Pump"

_membranes, 2022, doi:10.3390/membranes12111114_

Round 1

Reviewer 1 Report

 “Dense electrode layers supported microtubular oxygen pump” authored by Alexey Nikonov et al studied using a dense electrode layer supported microtubular oxygen pump. The work is interesting and in overall is well done, but the following comments should be carefully considered and make improvements in the revision.

1.   It is fundamental request for a porous electrode than the dense one because gas diffusion process demands a certain porosity. It is, however, interesting as reported:” the polarization resistance of the dense electrodes is greatly lower (by 2.5-5 times) than one of standard porous LSCF electrodes. Does this will affect the oxygen pumping efficiency? Please give a comprehensive discussion about this aspect.

2.   In introduction: In particular, “such electrolyte material 54 is Gd2O3 or Sm2O3 doped CeO2 (GDC or SmDC)”, Please add a few latest references here, particularly those related to low-temperature materials:, e.g.

[1] Shim J H. et al, Ceramics breakthrough [J]. Nature Energy, 2018, 3(3): 168-169.

[2] Gorte R. et al, Cooling down ceramic fuel cells [J]. Science, 2015, 349(6254): 1290.

[3] Bin Z. et al, A nanoscale perspective on solid oxide and semiconductor membrane fuel cells: materials and technology [J]. Energy Materials, 2021, 1:100

3.   This sentence of “In this works, systems that differ in both electrode and electrolyte materials were studied. Moreover the thickness of the dense layers did not exceed several microns.” Should be rephrased.

4.   About:” In this work the effect of dense composite layer on the electrode characteristics has 81 been investigated” A dense composite layer on the electrolyte introduces additional gas transfer resistance. Please explain the reason employing a dense structure instead of porous electrode.

5.    In “Materials and Methods” These info:” GDC has a cubic fluorite type structure with lattice 98 parameter a=5.424 Å. LSCF has a perovskite type structure (space group R-3c) with lattice parameters a=5.501 and c=13.407 Å. According to BET data, the LSCF particle size is 100 0.31 µm while the GDC powder consists of sphere-like shape particles with average size 101 of 10 nm….” These description should be transferred to Results and Discussion part.

6.   Please provide the full name of LC/C_2/3.

7.   In Results: About Figure 3 Please provide the standard unit of the scale bar.

8.   About Figure 12, the repeats in thermal cycling test should be at least 30 times.

9.   In Discussion: In the last paragraph:” in this paper has the lowest internal resistance and, therefore, has the lowest energy consumption for oxygen production (2.3 kW∙h∙m-3 347 )” The factors that have been taken into account when estimating cost should be clearly explained, as well as the general trend between energy cost and temperature. In addition, this oxygen pump technology should be compared with spontaneous oxygen permeation using MIEC membrane.

Reviewer 2 Report

Authors report experimental investigations on “Dense electrode layers supported microtubular oxygen pump”. The work is well presented in the manuscript. It can be accepted after following minor revisions:

1.       There are few typographical and grammatical errors in the manuscript. Author should check and correct these errors. (Examples: Line 14 & 15 of page No. 1 & Line 54 of Page No. 2). Few spelling mistakes are also observed. (In Line No. 126 of page no. 3, the word “simmetrical” should be “symmetrical”)

2.       Author should ensure the proper referencing for few statements. (Example: Line no. 78 & 79 should be supported by some reference.)

3.       Author should critically discuss few reports available on “Heterostructured solid electrolytes for OP applications” and “LSCF-GDC composite”.

4.       Company name and it origin for all the precursors should be mentioned in section 2.

5.       Authors claim that the LSCF, GDC and their composites are in single phase. Presentation of XRD data and associated discussion of all the synthesized powders in the manuscript is strongly recommended.

6.       Equivalent circuit (Randle’s circuit) associated with the impedance spectra and associated parameters should be mentioned and discussed for the better comparison of the compositions.

7.       The scale values of Fig. 4, 5 & 8 are not appropriately presented. Authors should follow the standard ways of presentation of the impedance data. [Ref: https://doi.org/10.3390/membranes11040289]

Reviewer 3 Report

It is not clear what the conclusions are for this work as there is no conclusion section. The main argument made in the introduction is that oxygen pumps (OPs) have not been adapted to medical use due to the long start-up time. Why would medical facilities need OPs? It is true that OPs can produce higher purity of oxygen (99%+), but is that high purity really needed? Today pressure swing absorption plants are widely available to cover higher oxygen demand in larger health facilities that can produce 90-95% purity of oxygen. The main result of this work are that a power consumption number in the developed design, which is not compared with any other widely adopted technologies, so it is hard to judge what does this number mean. While the research might be interesting to the oxygen pump community, the reviewer is not sure if this work meets the impact requirement of this journal. 
